# The role of human factors in paramedics' clinical judgement – A modified Delphi study

Anna Poranen[1]*, Anne Kouvonen[2,3], Hilla Nordquist[1,2,4]

1 Faculty of Medicine, University of Helsinki, Helsinki, Finland, 2 Faculty of Social Sciences, University of Helsinki, Helsinki, Finland, 3 Centre for Public Health, Queen's University Belfast, Belfast, Northern Ireland, 4 South-Eastern Finland University of Applied Sciences, Kotka, Finland

* anna.e.poranen@helsinki.fi

## Abstract

### Objective

In emergency medical services (EMS), work environments and circumstances are variable and make clinical judgement challenging. Little is known about the human factors that can affect paramedics' clinical judgement in different situations. The aim of this study was to identify the key human factors that can affect paramedics' clinical judgement during EMS missions, according to paramedic experts.

### Methods

A three-round modified Delphi study was conducted in 2024. Advanced level paramedics with at least five years of experience in EMS across Finland were included in the study. In the first two rounds, experts evaluated statements using seven- and five-point Likert scales. In Round 3, experts selected the five most significant human factors affecting paramedics' clinical judgement and describe related situations. The consensus level was set at ≥ 75%, and open-ended responses were analysed using thematic analysis.

### Results

Forty-seven experts participated in Round 1 and the response rates for Rounds 2 and 3 were 91% and 58%, respectively. After Round 1,16 statements were removed, 20 new statements were introduced, and four original statements were revised for Round 2. As consensus was not reached in the first two rounds, the analysis approach was modified to report medians and interquartile ranges (IQR). The experts identified 13 key human factors (IQR ≤ 1), with the four most significant being: 1) Unsafe locations involving a perceived risk to one's own and/or colleagues' work safety; 2) One's own attitude towards work tasks; 3) Support received from a

**Data availability statement:** The datasets generated and/or analysed in the current study are not publicly available because of privacy concerns. The informed consent indicated that the collected data would be used solely for this study, and no prior consent was obtained from the Delphi panelists to share their responses beyond the research team. While the content of the questionnaire and the response options are described in this paper, full transcripts of the open-ended responses have not been provided in order to preserve participant confidentiality. However, data access requests may be made to the corresponding author (anna.e.poranen@ helsinki.fi) upon reasonable request, provided that participant confidentiality can be maintained. Alternatively, requests may also be directed to the University of Helsinki Research Ethics Committee in the Humanities and Social and Behavioural Sciences (eettinen-toimikun ta@helsinki.fi).

**Funding:** Open access funded by Helsinki University Library. The funder had no role in study design, data collection and analysis, decision to publish, or preparation of the manuscript.

**Competing interests:** The authors have declared that no competing interests exist.

physician when carrying out the mission, and 4) One's own and one's work partner's problem-solving ability in relation to the situation at the scene.

## Conclusions

The present study found consensus on the four most significant human factors affecting paramedics' clinical judgement. Organisations need to provide adequate resources for situations involving occupational safety risks and provide support for paramedics' mental wellbeing.

## Introduction

Paramedics work in challenging circumstances, under time pressure, and have to treat and deal with a wide range of medical and social problems and make quick and effective decisions [1,2]. In many rapidly ageing societies, including Finland, emergency medical services (EMS) have increasingly started to complement other health and social care services, and paramedics often assess and treat patients in their homes in a more comprehensive way [3,4]. This all requires multidimensional and extensive decision-making skills and knowledge [3,4].

Clinical decision-making in EMS is a complex and multifaceted process that requires paramedics' to creatively adapt to variable environments [2,5,6]. There are several terms used to describe parts of the decision-making process, such as "*clinical reasoning*" and "*clinical decision-making*". Clinical reasoning can be defined as a cognitive process involving a paramedic first evaluating the problem. When a paramedic reaches a conclusion or makes a decision, the process is referred to as clinical decision-making [1]. The final outcome of this overall process, which also requires medical knowledge and skills, is a clinical judgement [1], and this term is used to describe the decision-making process in this study. In the clinical judgement process, paramedics gather information from multiple sources and situations: en route, on scene and post scene. Paramedics must recognise and analyse cues from environmental, event-specific and patient sources, specify hypotheses, solve a clinical problem, and evaluate the outcome all over again. In all of these phases, contextual factors affect information processing and must be acknowledged [1,6,7].

Various human factors influence paramedics' work in EMS settings [8]. Human factors refer to individual characteristics, as well as environmental and organisational aspects that interact with paramedics within the EMS setting [8,9]. It has been recognised that human factors also challenge paramedics' clinical judgement [8]. By understanding human factors, it is possible to enhance system performance, improve patient safety and paramedics' well-being and occupational safety [10–12].

Previous studies have demonstrated how paramedics and EMS clinicians make decisions and how safe paramedics' non-conveyance decisions are in EMS settings [2,4,6]. In addition, a few studies have focused on decision-making processes for specific patient groups and situations, for instance cardiac arrest, resuscitation and trauma [13,14]. Gugiu et al. [1] have also established a theoretical framework for

clinical judgement in EMS. However, studies focusing on the human factors that can affect paramedics' clinical judgement in different situations are lacking.

In this study, the aim was to identify the key human factors that affect paramedics' clinical judgement during EMS missions, according to paramedic experts.

## Materials and methods

### Study design

This was a modified Delphi study. The Delphi method is suitable for studies investigating complex issues where existing knowledge is incomplete or uncertain and standard research methods can be impractical. The Delphi method allows a panel of experts to reach a consensus on a specific subject in an iterative process [15,16]. Clinical judgement in EMS settings is a complex phenomenon, thus seeking consensus on the key human factors among paramedics using the Delphi technique is a suitable method for the study aim. The reporting recommendations "*Delphi studies in social and health sciences–recommendations for an interdisciplinary standardized reporting"* (DELPHISTAR) were used to guide the reporting of this study [17].

### Setting

In Finland, EMS is organised by 21 wellbeing services counties (health and social care regions) and the Helsinki and Uusimaa Hospital District Joint Municipal Authority (HUS) Group, which together form five collaborative areas; Northern, Eastern, Inland, Western and Southern Finland [18,19]. Each wellbeing services county has at least one EMS field supervisor who is an advanced level paramedic with extensive work experience of EMS and sufficient administrative and operational expertise. In Finnish EMS, paramedics are classified into two levels: advanced level (comparable to advanced paramedic) and basic level (comparable to emergency medical technicians). Advanced level paramedics hold a bachelor level emergency care degree (with integrated registered nursing competencies) or registered nursing degree with additional advanced level prehospital specialisation course. Basic level paramedics are practical nurses with prehospital specialisation, firefighters or registered nurses [20,21]. At the time of the study, the Finnish EMS consist of seven helicopter emergency medical units (HEMS) and several ground-based EMS physician units. Both types of units frequently provide care instructions to paramedics over the phone, but they also take part in specific high-risk emergencies where their involvement is considered beneficial [22].

### Study population

The study participants (hereinafter: experts) were recruited via social media. The inclusion criteria were that experts must be advanced level paramedics, with at least five years of work experience in EMS and currently working in EMS in Finland. A recruitment ad was posted (Jan 2024) on the Finnish Facebook group "Ensihoidon uutiset" (in English: Prehospital EMS News) which has almost 5000 members across Finland. The administrator of the group was contacted beforehand to get permission to publish the recruitment ad.

Potential experts were invited to participate in the study via a REDCap (Research Electronic Data Capture tool, hosted by the University of Helsinki [23]) link, which allowed them to get more information of the study and access the data protection statement. If a potential expert was interested in participating in the study, they clicked a consent to participate box. After that, they were asked to fill out their background information (age, gender, length of work experience, advanced or basic level paramedic, and health and social care collaborative area that they represented; Northern, Eastern, Inland, Western or Southern Finland). If the participant met the inclusion criteria, they were asked to provide their email address to receive a REDCap link for the first questionnaire via email. If the potential participant did not meet the inclusion criteria, they were excluded, and their information was deleted.

## Ethical considerations

Approval from the University of Helsinki Research Ethics Committee in the Humanities and Social and Behavioural Sciences was granted on 13th of November 2023 (Statement 74/2023). Participation in the study was voluntary. All the participants provided written informed consent after having had an opportunity to review a study description and data protection statement via REDCap. The experts' anonymity was ensured throughout the study. Their personal data was accessible only to the first author (AP) upon participation confirmation via REDCap, and questionnaire links were sent via blind copy to protect email addresses.

## Data collection and analysis

In the first part of the study, a literature review was conducted to identify relevant human factors in EMS settings [2,6–8,13,14,24–29]. The literature review process is shown in the S1 File. Key factors affecting paramedics' clinical judgement were extracted from previous studies and statements for the Round 1 questionnaire were developed from them. The questionnaire was piloted with a group of experienced advanced level paramedics (n = 17) who were studying in a master's programme in the South-Eastern Finland University of Applied Sciences to ensure that all statements were understandable and unambiguous. Pilot participants highlighted a statement if it was unclear or not understandable. Unclear statements were revised by the authors. The online questionnaires were delivered for experts to respond using the secure and unique REDCap link.

### Round 1

In Round 1, the identified human factors were classified under nine main categories: *Organisational factors, Work circumstances, External factors, Team & pair work, Experience, Cognitive factors, Workload and stress, Work motivation and Personal factors.* The experts evaluated a total of 44 statements regarding human factors and were asked to indicate how much they believe each might affect their clinical judgement. The response options were presented using a seven-point Likert scale where 1 represented "not at all" and 7 "very much". There was also an option 8, "do not know". Furthermore, an open-ended question was included after each main category to give the experts an opportunity to provide any further thoughts or highlight missing issues regarding human factors that might affect paramedics' clinical judgement in EMS settings. At the beginning of the questionnaire, experts were provided with definitions of what is meant by human factors and clinical judgement in the context of this study.

Before data collection, the consensus criterion was defined as at least 75% agreement in the categories of "much" and "very much" (Likert scale 6 or more). This is a common threshold in Delphi studies [16]. It was also decided that if 20% or more of the experts selected the statement as having "very little" or "not at all" (Likert scale 2 or less) impact on their clinical judgement, it was counted as disagreement and the statement was removed from subsequent Delphi rounds [30]. The online questionnaire was open for two weeks (from 24 Jan to 7 Feb 2024). After one week a reminder was sent to those experts who did not initially respond.

Descriptive statistics were calculated using IBM SPSS Statistics for Windows, version 28.0 (IBM Corp., Armonk NY) and the analysis was conducted in February 2024. For the analysis, the 7-point Likert scale was converted to a 5-point Likert scale where options 1 and 2 were merged to form the option 1 = "not at all" and options 6 and 7 were merged to form the option 5 = "very much". Frequencies, levels of consensus and mean values were calculated, and the demographics of the experts were collected. When calculating the mean values, option 8 ("do not know") was excluded. Open-ended answers were analysed using the principles of thematic analysis and were pseudonymised, meaning that no personal information was included. Pseudonymisation was conducted by the first author before the units of meaning were shared with the other authors. All the new aspects that were suggested by an expert in answers to open ended questions in Round 1 were categorised under the original categories and formulated as statements for Round 2.

## Round 2

The Round 1 analysis provided inputs for Round 2 and the second questionnaire was sent anonymously to the panel for review. Following the Delphi method, experts were given the results of Round 1 in the cover letter so they had the opportunity to see the distributions of their answers [15,17]. Round 2 included 20 new statements, four original statements were revised, and thus a total of 48 statements were evaluated by the experts.

The experts were asked to evaluate the statements on the following 5-point Likert scale: 1 = "not at all", 2 = "a little", 3 = "neutral", 4 = "much", and 5 = "very much". An additional option 6 "do not know" was also available and this round did not include any open-ended questions. The online questionnaire was open for two weeks (from 21 Feb to 6 March 2024) and after one week a reminder was sent to those experts who did not initially respond.

While the initial plan was to analyse consensus based on percentage agreement, the analysis approach was adjusted after Round 2 and medians and interquartile ranges (IQR) were calculated allowing a more nuanced assessment of consensus in Likert-scale responses. After Round 2, the statements that received the highest ratings on the Likert scale from the experts, which in this study were categorised as "much" or "very much," were considered to have the highest level of agreement [31]. If the IQR was 1 or less, consensus was obtained [32].

## Round 3

In Round 3, the experts who participated in Round 2 received their third online questionnaire. After the Round 2 analysis, 13 statements reached IQR 1. In Round 3, experts were asked to highlight the five most significant human factors out of 13 that might affect their clinical judgement in EMS settings. For each highlighted statement, the experts were asked to describe in what kind of clinical judgement situation the said human factor comes into effect. The online questionnaire was open for two weeks (from 7 to 21 April 2024). After one week a reminder was sent to those experts who did not initially respond, and a second reminder was sent five days before the deadline.

In the analysis (May–Oct 2024), the highlighted statements were assigned a score from five to one points with the most significant assigned five points and the fifth most significant assigned one point. Then the total score of each statement were calculated. Open-ended responses were analysed using thematic analysis, following the six phases of thematic analysis described by Braun and Clarke [33]. Prior to the analysis, the experts' comments were grouped according to the statements they related to. Four human factors received a higher total score than the others, indicating that the experts considered them to have the most significant impact on paramedics' clinical judgement; only the comments related to these four human factors were analysed using thematic analysis. The first author started the analysis by familiarising herself with the data by reading the comments several times, searching for meanings and making notes on initial ideas for themes. Then coding began without the use of any software. Data extracts were collated under the same codes for possible themes. In the third phase, different codes were sorted into subthemes and then main themes were formed. Once the first draft of the themes was created, the themes were reviewed by the other authors, refined and overlapping themes were revised. Finally, a thematic map of each human factor was formulated.

## Results

### Round 1

The first round involved 47 experts, of whom 25 were women and 22 men. The majority of the experts were aged 30–39, and nearly half had 10–15 years of experience in EMS. The demographic characteristics of the experts from all rounds are presented in Table 1.

The highest level of agreement was 68% for the statement *"Unsafe locations involving a perceived risk to one's own and/or colleagues' work safety"* but none of the statements reached consensus. Open-ended questions resulted in a total of 44 individual comments which were used to form 20 new statements. Four original statements were revised based on

**Table 1. Demographic characteristics of the experts.**

| Characteristic | | Round 1 n | Round 2 n | Round 3 n |
|---|---|---|---|---|
| Gender | Female | 25 (53.2) | 22 (51.2) | 18 (66.7) |
| | Male | 22 (46.8) | 21 (48.8) | 9 (33.3) |
| | Other | 0 | 0 | 0 |
| | Prefer not to say | 0 | 0 | 0 |
| Age | < 30 | 5 (10.6) | < 5 | < 5 |
| | 30-39 | 22 (46.8) | 20 (46.5) | 13 (48.1) |
| | 40+ | 20 (38.5) | 19 (44.2) | 11 (40.7) |
| Work experience | < 10 years | 12 (26.1) | 13 (30.2) | 7 (25.9) |
| | 10-15 years | 23 (48.9) | 19 (44.2) | 14 (51.9) |
| | > 15 years | 12 (26.1) | 11 (25.6) | 6 (22.2) |
| Collaborative area | Northern Finland | 9 (19.1) | 7 (16.3) | 5 (18.5) |
| | Eastern Finland | 6 (12,8) | 6 (14,0) | < 5 |
| | Inland Finland | 9 (19.1) | 8 (18.6) | 7 (25.9) |
| | Western Finland | 6 (12.8) | 5 (11.6) | < 5 |
| | Southern Finland | 17 (36.2) | 17 (39.5) | 10 (37.0) |

the comments, for instance, *"Lack of experience"* was revised to *"Paramedic's self-assessment that they lack experience"*. As 20% or more participants had selected the "not at all" or "a little" answer option, 16 statements were not included in subsequent rounds.

## Round 2

In Round 2, the response rate was 91% (n = 43). The required level of consensus (75% or more) was not achieved for any of the statements. Thus the analysis strategy was modified. The medians of the statements were calculated, and the median of 30 statements was 4, with the IQR of 13 of these statements being ≤1. The results of Rounds 1 and 2 are presented in Table 2.

## Round 3

In Round 3, 27 experts (response rate 58%) participated. In this final round, the number of women participants was higher (n = 18, 66.7%), but the age distribution and years of work experience were almost the same as in the previous two rounds (see Table 1).

Four key human factors influencing paramedics' clinical judgement clearly stood out when the total scores were calculated (Fig 1). The highest scoring factor (with a total of 68 points) was *"Unsafe locations, involving a perceived risk to one's own and/or colleagues' work safety"* which had also emerged as the most significant factor in the first two rounds. The second most significant human factor (*"One's attitude towards work tasks"*) also scored 19 points higher than the third. The third and forth most significant human factors affecting paramedics' clinical judgement were related to team and pairwork, as well as cognitive factors, and received a total of 39 points (*"Support received from a physician when carrying out the mission"*), and 36 points (*"One's own and one's work partner's problem-solving ability in relation to the situation at the scene"*), respectively.

## Situations in which key human factors come into effect

In the qualitative thematic analysis, the aim was to describe the clinical judgement situations in which the most significant human factors come into effect. The main themes and subthemes of each human factor are displayed in Fig 2.

**Table 2. The results of Rounds 1 and 2.**

| Main categories | Statements | Round 1 | | | Round 2 | |
|---|---|---|---|---|---|---|
| | | Mean value | Level of consensus (%)[a] | Excluded after Round 1[b] | Median | IQR |
| **Organisa-tional factors** | Lack and/or ambiguity of guidelines | 4.49 | 36 | | 4 | 2 |
| | Inadequacy of training provided at the organisational level | 3.80 | 23 | X | — | — |
| | Lack of organisational level support if an adverse event occurs | 4.93 | 28 | | 4 | 1.25 |
| | Lack of feedback (limited opportunities to reflect on one's actions and learn) | 4.85 | 43 | | 4 | 2 |
| | Lack of support and trust from supervisor/employer | 3.89 | 23 | X | — | — |
| | Lack of rules and regulations defined by supervisor/employer | 3.74 | 19 | X | — | — |
| | Negative attitude by the physician consulted towards requests for treatment instructions | [c] | | | 4 | 2 |
| | Emergency medical care legislation's ambiguities and diverse interpretations | [c] | | | 3 | 2 |
| | Change of organisation and resulting changes | [c] | | | 2 | 1 |
| | Insufficient number of emergency units in the operational area | [c] | | | 4 | 2 |
| | Lack of values defined by the supervisor/employer that influence work (e.g., whether the right treatment decision is more important than client satisfaction) | [c] | | | 3 | 1 |
| | Even distribution of workload by EMS field supervisor | [c] | | | 3 | 2 |
| | Lack of peer support in the occurrence of an adverse event | [c] | | | 3 | 2 |
| | No room for paramedic-led situational assessment and action due to the organisation's defined protocols | [c] | | | 3 | 2 |
| **Work circum-stances** | Nighttime | 4.15 | 19 | | 3 | 2 |
| | Mission at the end of a shift | 4.38 | 26 | | 4 | 1 |
| | Darkness | 3.85 | 15 | X | — | — |
| | Unsafe locations, involving a perceived risk to one's own and/or colleagues' work safety | 5.97 | 68 | | 4 | 1 |
| | Rapidly changing situations | 4.13 | 21 | | 3 | 2 |
| | Urgency | 3.77 | 17 | | 2 | 1 |
| | Time pressure | 3.83 | 9 | | 2 | 2 |
| | Noise | 3.26 | 9 | X | — | — |
| | Distance to the hospital | 4.77 | 43 | | 4 | 2 |
| | Weather conditions | [c] | | | 2 | 1 |
| **External factors** | Family members/bystanders (Round 2: Pressure and/or behavior of family members and/or bystanders) | 3.72 | 17 | | 2 | 1 |
| | Pets at the scene | 2.38 | 9 | X | — | — |
| | Conditions of the patient's/client's home or location | [c] | | | 3 | 2 |
| **Team & pair work** | Lack of situational awareness (on the part of work partners and/or team members) | 5.02 | 38 | | 4 | 1 |
| | Misunderstood communication | 5.17 | 51 | | 4 | 1 |
| | Lack of communication | 5.11 | 49 | | 4 | 2 |
| | Lack of trust | 5.23 | 49 | | 4 | 2 |
| | Challenges in co-operation | 4.91 | 34 | | 4 | 2 |
| | Disagreement within the team participating in the mission (Round 2: Disagreement between work partners/within the team participating in the mission) | 5.04 | 43 | | 4 | 1 |
| | Support received from the partner and/or the team when carrying out the mission | [c] | | | 4 | 1 |
| | Support received from a physician when carrying out the mission | [c] | | | 4 | 1 |

*(Continued)*

**Table 2.** (Continued)

| Main categories | Statements | Round 1 | | | Round 2 | |
|---|---|---|---|---|---|---|
| | | Mean value | Level of consensus (%)[a] | Excluded after Round 1[b] | Median | IQR |
| Experience | Intuition | 4.87 | 30 | | 4 | 1 |
| | Lack of experience (Round 2: Paramedic's self-assessment that they lack experience) | 5.06 | 53 | | 4 | 1.5 |
| Cognitive factors | Misleading information about the mission from the emergency response centre | 3.13 | 6 | X | — | — |
| | Information overload | 3.79 | 11 | | 3 | 2 |
| | One's thoughts are somewhere else (holidays, personal issues/challenges in personal life) | 2.83 | 4 | X | — | — |
| | Preconceptions | 4.19 | 15 | | 3 | 2 |
| | Ability to quickly gather necessary information during an emergency task | [c] | | | 4 | 1 |
| | One's own and one's work partner's problem-solving ability in relation to the situation at the scene | [c] | | | 4 | 1 |
| Workload and stress | Tasks that challenge one's professionalism and/or expertise | 4.43 | 26 | | 4 | 1 |
| | Mental pressure/mental workload | 4.09 | 23 | | 3 | 2 |
| | Past events linger in the mind | 2.21 | 2 | X | — | — |
| | Insufficient recovery between shifts | 3.57 | 11 | X | — | — |
| | Exhaustion/burnout | 4.20 | | X | — | — |
| | Accumulation of work-related stress and/or workload | [c] | | | 4 | 2 |
| | No opportunity for breaks during the shift | [c] | | | 4 | 1 |
| Work motivation | One's attitude towards work tasks | 5.21 | 40 | | 4 | 1 |
| | Frustration (Round 2: Frustration with the current situation) | 4.83 | 45 | | 4 | 1.25 |
| | An increasing number of non-urgent EMS missions | 3.96 | 34 | X | — | — |
| | Frequent visits to the same client/patient | [c] | | | 4 | 2 |
| | Work partner's attitude and motivation | [c] | | | 4 | 2 |
| | Meaningfulness of missions and/or their relevance to emergency medical care | [c] | | | 4 | 2 |
| Personal factors | Hunger | 3.57 | 17 | X | — | — |
| | Fatigue | 4.49 | 19 | | 4 | 2 |
| | Age | 2.67 | 2 | X | — | — |
| | Feeling of empathy | 4.50 | 28 | | 3 | 2 |
| | Too much selflessness and altruism towards patients | 4.27 | 17 | | 3 | 2 |
| | Powerlessness (the feeling of being unhelpful) | 3.18 | 2 | X | — | — |
| | Feeling of inadequacy | 3.15 | 2 | X | — | — |
| | Feeling that the work of paramedics is not valued | [c] | | | 4 | 2 |

[a]At least 75% agreement in the categories of "much" and "very much" (Likert scale 6 or more).

[b]At least 20% agreement in the categories of "very little" or "not at all" (Likert scale 2 or less).

[c]Added after Round 1.

## Unsafe locations, involving a perceived risk to one's own and/or colleagues' work safety

Under the theme *"Unsafe locations, involving a perceived risk to one's own and/or colleagues' work safety"* the main theme *"The choice of tactics for managing the mission"* describes situations in which paramedics need to make a decision about the tactics for managing a mission. Its four subthemes are: *"Vigilance"*, *"Time management"*, *"Putting your own safety before the patient's best interests"* and *"Not going to the scene"*.

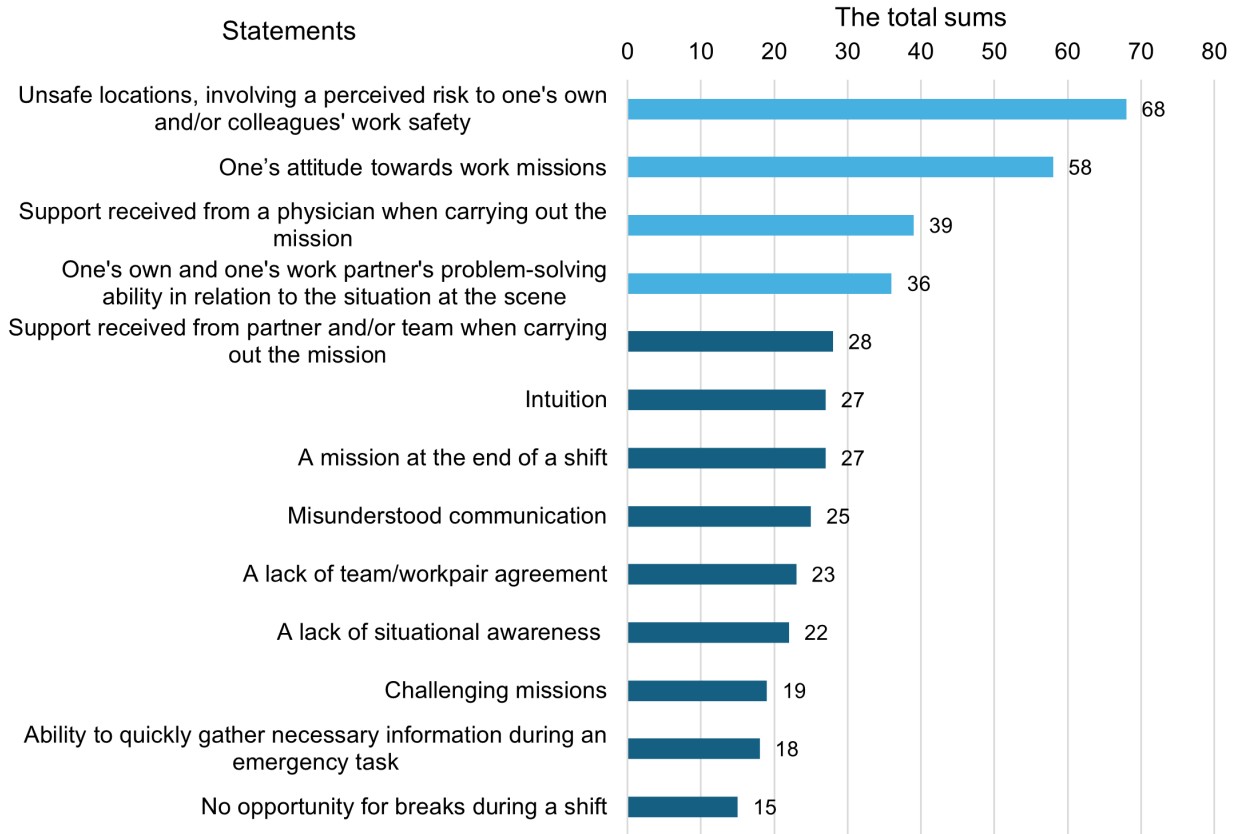

**Fig 1. The results of Round 3; the total score of each statement.**

According to the experts, the subtheme *"Vigilance"* plays a key role in these unsafe situations and clinical judgement becomes more difficult when attention is affected if the situation is perceived as threatening to work safety in any way. In such cases, focus shifts to the environment, and task management may be disrupted or even interrupted. In addition, clinical judgement can become limited to merely surviving the situation.

In unsafe work environments, clinical judgement also has the subtheme *"Time management"*, as there is pressure to either make decisions very quickly or delay patient care due to perceived or actual threats to occupational safety. The challenges in clinical judgement are also reflected in the time spent on patient examination and the thoroughness of the assessment. In these situations, something may be overlooked.

*"An unsafe situation creates pressure to make decisions quickly in order to leave the scene. Feeling unsafe also disrupts clinical judgement, and there is a risk that something important may be overlooked." Expert 26 = E26*

The subtheme *"Putting your own safety before the patient's best interests"* refers to situations where paramedics leave the scene for their personal safety, leaving the patient behind. It also involves assessing the safety of oneself and one's colleagues, such as when police are not involved on the same EMS mission.

*"In potentially threatening situations, one's own and one's colleagues' safety takes priority on the value scale, with patient care following after that." E16*

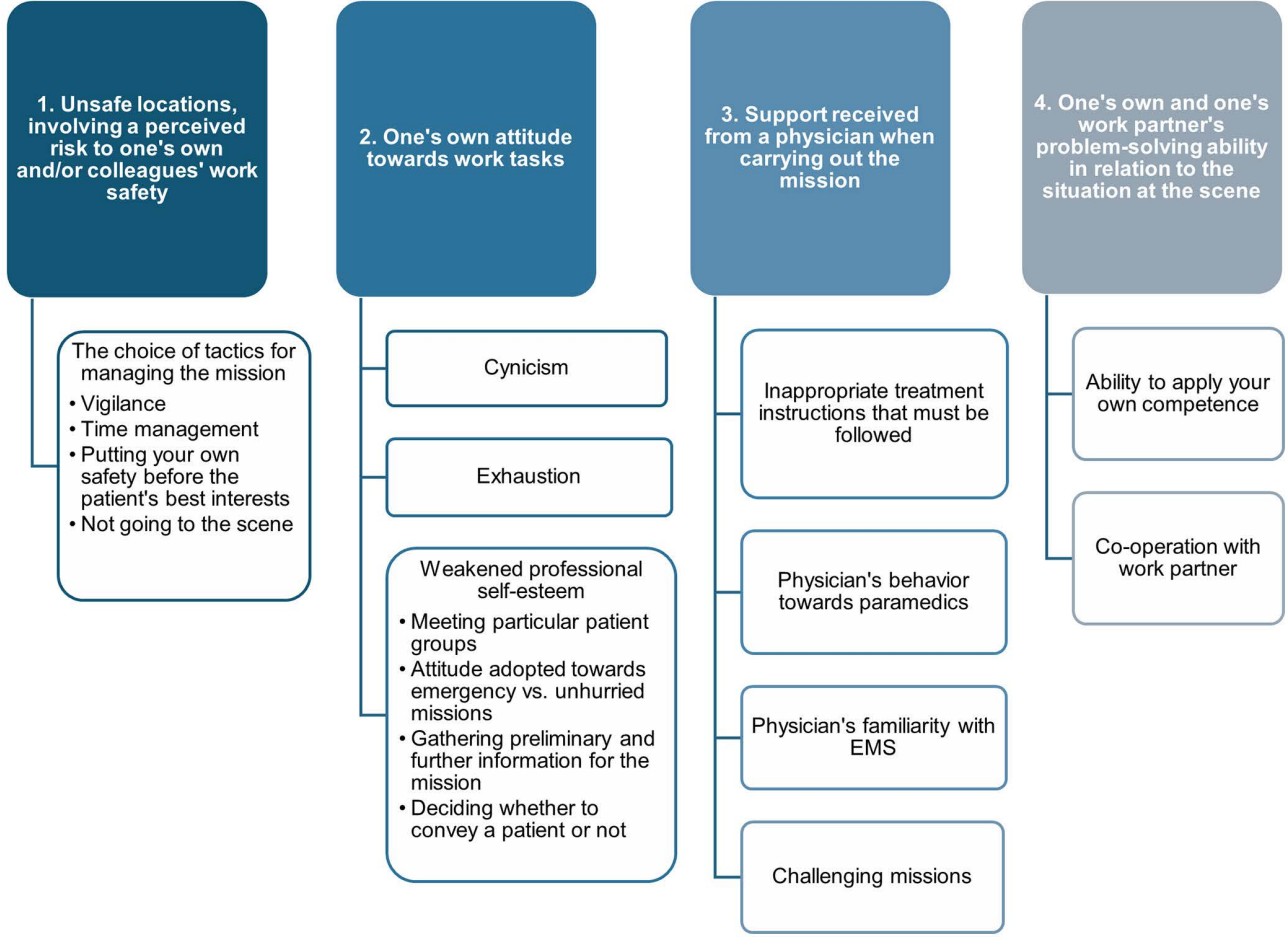

**Fig 2. Clinical judgement situations in which human factors come into effect in EMS settings.**

The subtheme *"Not going to the scene"* reflects situations where tactics change, paramedics consider whether to approach the scene at all and if so, how to work there. According to the experts, an experience of insecurity alters their attitude towards the mission due to the perceived safety threat, making the primary goal to leave the scene. Experts' responses revealed that paramedics may decide to transport the patient to the hospital, even with limited information, simply because they want to leave the scene quickly.

*"It is difficult to draw a clear line between when a situation poses a real danger or just a threat. However, if an employee feels threatened, afraid, or is on high alert, it significantly impacts clinical judgement, starting with whether to enter the location or not."* E24

**One's own attitude towards work tasks**

*"One's own attitude towards work tasks"* has three main themes: *"Cynicism"*, *"Exhaustion"* and *"Weakened professional self-esteem"*. The last theme has the following four subthemes that reflect situations in which weakened professional self-esteem emerges in clinical judgement, *"Meeting particular patient groups"*, *"Attitude adopted towards emergency vs.*

unhurried missions", "Gathering preliminary and further information for the mission", and "Deciding whether to convey a patient or not".

According to the experts, the main theme "Cynicism" causes a negative attitude towards patients, and clinical judgement changes when paramedics do not take all matters seriously. If a paramedic has a cynical attitude, decisions are often made hastily, and issues are not investigated thoroughly enough.

Regarding the main theme "Exhaustion", the experts pointed out that the attitude of paramedics becomes evident in situations where they experience fatigue, indifference, and a sense of urgency. In such cases, the ability to recognize the limits of their own expertise is also reflected in their attitude, which in turn affects clinical judgement.

*"Fatigue, irritability, indifference, urgency, and the ability to admit when something is not known." (E11)*

The main theme "Weakened professional self-esteem" reflects situations where a paramedic's diminished professional self-esteem becomes evident through their attitude. The subtheme "Meeting particular patient groups", describes how paramedics' attitudes towards so-called "frequent callers", mainly mental health and substance abuse patients, as well as patients from ethnic minority backgrounds, affect how patients are listened to, how thoroughly they are examined, or whether they are examined at all. The responses indicate that some paramedics unfortunately have a particularly negative attitude towards patients with substance abuse problems, and certain issues may easily go undiagnosed when encountering a so-called "frequent caller" who repeatedly complains of the same symptoms. Often, the approach to the task may already be influenced by the assumption that it is nothing serious, and that the complaint is the same as before, which can easily guide clinical judgement.

*"Especially with so-called "frequent callers," patients with substance abuse and mental health issues, or those from ethnic minority backgrounds, the paramedic's attitude has a significant impact on clinical judgement. Are the symptoms taken seriously? Is the patient listened to? Are they examined as thoroughly as usual, or is there reluctance to investigate at all?" E2*

Regarding in the subtheme "Attitude adopted towards emergency vs. unhurried missions", the experts pointed out that the attitude towards work is reflected in attitudes towards people and their situations, particularly in how paramedics approach urgent versus non-urgent missions: is their attitude the same, and how are patients valued in these tasks? According to the experts, especially in tasks where the patient's social factors are involved, the paramedic's attitude towards their work and their view of their own role in relation to the core mission of EMS can significantly influence clinical judgement.

The subtheme "Gathering preliminary and further information for the mission" refers to how the attitude a paramedic has when heading out to a call influences the preconceived approach they take towards the task, according to the experts. This preconceived attitude strongly affects how tasks are handled in EMS as well as the paramedic's clinical judgement. If the paramedic has already decided what the situation is or expects a certain outcome, it can lead to the task being misdirected or handled incorrectly.

*"When preparing for each task, one's attitude already influences initial mindset (e.g., dismissing it as a "false alarm," expecting a quick response, or anticipating a real emergency). This preconceived attitude can significantly affect how the task is handled and the decisions made, especially if one has already mentally decided what the situation is before arriving. It can lead to a failure to see the bigger picture." E20*

The subtheme "Deciding whether to convey a patient or not" reflects situations where paramedics' decisions regarding the need to transport a patient are influenced by their own attitude or lack of motivation.

*"I have noticed that a lack of motivation or other negative attitudes in work partners has affected their clinical judgement, for example, in determining the need for transport." E27*

### Support received from a physician when carrying out the mission

*"Support received from a physician when carrying out the mission"* has four main themes that reflect situations where paramedics receive support from a physician, and whether it is sufficient or inadequate, to assist in their clinical judgement. The four main themes are as follows: *"Inappropriate treatment instructions that must be followed"*, *"Physician's behavior towards paramedics"*, *"Physician's familiarity with EMS"*, and *"Challenging missions"*.

According to the experts, if a paramedic does not receive the desired medical assistance from a physician, or if the physician's treatment instructions, such as regarding patient transportation to a hospital over long distances, are not valid for the specific situation, these issues can affect the paramedics' clinical judgement. Additionally, the experts stated that paramedics are put in a difficult position when it comes to clinical judgement if they do not receive any treatment instructions from the physician at all, or if the instructions provided are of poor quality.

*"If, for example, a consultation cannot be obtained or its quality is poor, this naturally places the paramedic in a challenging position when it comes to clinical judgement." E22*

Regarding the main theme *"Physician's behaviour towards paramedics"*, the experts emphasised the challenges posed by certain behaviors. The experts highlighted that if a physician does not listen, is dismissive of requests for treatment instructions, or dominates and overrides the paramedic's suggestion regarding transportation needs, clinical judgement becomes even more challenging.

Regarding the main theme *"Physician's familiarity with EMS"*, experts mentioned that when a paramedic has to ask for treatment instructions from a physician who is not familiar with prehospital emergency care, they may find themselves alone in the clinical judgement situation, as criticism comes instead of support.

*"If you can't reach the designated physician (FH [FinnHEMS], the on-call EMS physician) and have to consult with a regional hospital physician whose attitude toward prehospital care is dismissive, you might not receive any support for clinical judgement and end up feeling isolated in the situation. In the worst case scenario, you might even face harsh criticism." E5*

The experts found that clinical judgement becomes more difficult in particularly challenging situations, for example, when there is a lack of agreement with the patient and their family about the situation, or when, for some reason, the overall picture of the situation cannot be communicated to the on-call EMS physician. In such situations, paramedics felt that support from the physician involves offering reassurance and an objective perspective, which aid in resolving the uncertainty. On challenging missions, such as when treating acutely ill patients, support from a physician makes clinical judgement easier. However, the experts also pointed out that if the physician becomes involved in the challenging mission, the treatment plan may change, which in turn complicates the paramedics' clinical judgement.

*"A difficult decision, such as whether to leave a palliative patient without transport, becomes somewhat more bearable if the physician approaches the situation with humanity and collaboratively plans the best means of relief for the patient. Similarly, treating an acutely seriously ill patient feels less burdensome if the physician provides background support regarding treatment." E23*

### One's own and one's work partner's problem-solving ability in relation to the situation at the scene

*"One's own and one's work partner's problem-solving ability in relation to the situation at the scene"* has two main themes: *"Ability to apply your own competence"* and *"Co-operation with work partner"*.

The main theme *"Ability to apply your own competence"* involves problem-solving skills becoming apparent in situations that may be ambiguous and for which the paramedic has received no training or no treatment instructions are available. The experts' responses highlighted that prehospital missions are rarely focused on a single issue, so prioritisation is needed in clinical judgement, as well as the ability to apply and offer other relevant services available in one's working area. Problem-solving skills are especially important on non-urgent emergency missions or missions that involve social problems.

*"Prehospital tasks are rarely textbook examples of sudden illnesses or accidents, which makes the ability to apply one's knowledge and expertise crucial. Additionally, in many prehospital situations, social problems also need to be addressed, for which there may be no formal training or guidelines available." E2*

The main theme *"Co-operation with work partner"* describes situations where collaboration with one's work partner is significant for clinical judgement. According to the experts, the clarity of actions between the partners and situational awareness can impact clinical judgement, for example, in situations where one partner complements the other if something has been overlooked. On the other hand, problem-solving ability in collaboration with the partner also emerges when there are differences in perspective on how to handle a situation. Furthermore, the attitudes of partners influence the collaboration, as a negative attitude will be reflected in the way the partner approaches the patient's well-being.

## Discussion

The aim of this Delphi study was to identify key human factors that affect paramedics' clinical judgement during EMS missions from the point of view of paramedic experts. We found that there were four highly significant human factors. The overwhelmingly most significant human factor found to affect paramedics clinical judgement was *"Unsafe locations involving a perceived risk to one's own and/or colleagues' work safety".*

Missions that threaten occupational safety are common in EMS, and this emerged as a significant issue in a previous study by Nordquist and Kouvonen [34], in which Finnish paramedics reported experiencing both verbal and physical violence at work. Other previous studies [2,24] have reported similar findings regarding the relationship between clinical judgement and occupational safety risks as those of this study. Paramedics may decide to delay entering the scene because of the safety risks and assess the situation continuously which may change their previous decisions on how to handle the situation. It is noteworthy that in our study, feelings of insecurity and the risk to both one's own and one's work partner's safety stood out so clearly as the most significant factor. This is in line with previous studies showing that risks to occupational safety and the risk of violence against paramedics are common [35,36]. For example, systematic review by Murray et al. [35] shows that 57–93% of paramedics have experienced verbal and/or physical violence at least once during their career.

EMS involves a variety of stressors, for instance paramedics are exposed to traumatic and emotionally taxing situations, shift work, and work in challenging environments [24,37–39]. Burnout and stress among paramedics have been examined, and some studies suggest that there may be a connection between occupational safety risks and mental strain. In a study by Braun et al. [40], the results showed a small association between experienced violence and burnout among paramedics. Additionally, Ericsson et al.'s [39] study on the job demands of paramedics found that occupational safety risks increase paramedics' mental workload. Feelings of exhaustion, cynicism and inefficacy at work are characteristics of burnout, which result from unmanaged chronic work-related stress [41,42]. Our study noted these three characteristics in paramedics' attitudes, and this result indicates a link between burnout and clinical judgement but also highlights that emotional stability is needed to handle missions [24]. Our results suggest that attitude is also linked to the preconceptions a paramedic may have when heading to a mission, and can change the paramedic's actions in the situation, potentially leading, for example, to an incorrect decision. A similar result was found in a Canadian study by Reay et al. [2].

Paramedics have the opportunity to request support from a physician when making a clinical judgement and requesting treatment instructions from a physician is often required by a standard operating procedure. The findings of this study suggest that paramedics consider physicians' support an important factor in clinical judgement, but the quality and amount of support vary for various reasons. A previous Finnish study [39] suggested that a lack of support in clinical decision making contributes to cumulative stress and similar findings were uncovered in this study. Moreover, in our study paramedics felt that when an EMS physician joins a challenging collaborative mission, the operational tactics may change, which can make clinical judgement more difficult for them. A somewhat similar result was found in a study on Dutch paramedics' decision-making in traumatic cardiac arrest situations, revealing differences in operational tactics when the HEMS team joined the mission [26]. In line with another study [43], the treatment instructions given by EMS physicians were perceived as important and supportive of paramedics' work.

Paramedics encounter patients with a wide range of health issues in their work, and missions can be complex. Challenging calls also require good teamwork skills and the ability to make demanding decisions [24,28]. A lack of situational awareness and communication issues can pose a risk to patient safety, making it more difficult for paramedics to make decisions [28]. Similar results were found in our study, and the effectiveness of collaboration was considered an important factor from a clinical judgement perspective. However, as this study revealed, social distress, such as loneliness may also present on EMS missions and challenge paramedics' clinical judgement abilities as there may not be guidelines or training for these situations. Ericsson et al.'s [39] study noticed a similar issue. Complex missions and social distress increase the workload of paramedics, as they feel unable to care for patients as well as they would like because of limited knowledge. Nevertheless, more research is needed on handling social distress situations in EMS to develop a deeper understanding of the types of social distress paramedics encounter in the field.

## Strengths and limitations

For this study, the modified Delphi method was ideal to achieve the research aim, as paramedics' clinical judgement is a complex and under-researched phenomenon [44]. However, under the Delphi method, knowledge construction relies on the experts' current knowledge, and the reliability of the study therefore depends on the participating experts [45]. One of the strengths of this study is that there were an adequate number of experts in the first two rounds, as the optimal number of participants in a Delphi study is considered to be between 30 and 50 [15]. Most of the experts also had 10 years or more of experience in EMS and represented a broad range of collaborative areas across Finland. However, the number of respondents dropped remarkably in the final round, with the number of male experts in particular decreasing despite reminders being sent. These dropouts may have led to consensus bias and could be due to completing the survey being seen as too time-consuming or demanding in the first two rounds [46]. Another potential bias may be self-selection as participation may have predominantly included paramedics who were particularly motivated to share their views on the topic. This may have influenced the representativeness of the findings. Moreover, recruiting experts only from Finland may limit the broader generalisability and transferability of our results.

The consensus criteria were defined in advance, as is recommended in Delphi studies [45]. However, the decision to base consensus on the percentage of agreement was modified when analysing the Round 2 results because no consensus of 75% was found during the first two rounds. Thus, another consensus definition was used from then on and reported transparently to increase the validity of this study. A further strength of this study is that the questionnaire was piloted before Round 1 and modified based on the responses. Additionally, the experts were given the opportunity to review a summary of the results from Round 1, allowing them to re-think their responses in Round 2 [15,17,45].

In their open-ended answers in Round 1, the experts mentioned factors that, according to the literature, do not fully align with the definition of human factors [12,47]. Previous research has noted that paramedics find it challenging to understand and describe human factors in practice [8], and this was evident in this study as well. However, the experts were informed at the beginning of the questionnaire what was meant by human factors in this study.

## Conclusion

The present study found a consensus on the four most significant human factors affecting paramedics' clinical judgement and the findings also highlighted situations and actions in which these four factors come into effect. This study brings new perspectives to paramedics' clinical judgement, as to our knowledge, it is the first study to examine the human factors influencing clinical judgement from paramedics' points of view. The results can be applied in practice by ensuring that organisations provide adequate resources for situations involving occupational safety risks and provide support for paramedics' mental stability. Overall, organisations can use the findings of this study to offer sufficient training, support and tools for clinical judgement in the challenging conditions of EMS.

## Supporting information

**S1 File. Literature review process to identify relevant human factors in clinical judgement in EMS settings.**
(PDF)

## Acknowledgments

The authors would like to acknowledge all those Finnish paramedics who gave up their precious time to take part in this study.

## Author contributions

**Conceptualization:** Anna Poranen.

**Formal analysis:** Anna Poranen.

**Methodology:** Anna Poranen.

**Supervision:** Anne Kouvonen, Hilla Nordquist.

**Writing – original draft:** Anna Poranen.

**Writing – review & editing:** Anna Poranen, Anne Kouvonen, Hilla Nordquist.

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
