## [Decision Letter · Decision Letter 0]

26 Jul 2025

PONE-D-25-34423
The role of human factors in paramedics’ clinical judgement – A modified Delphi study
PLOS ONE

Dear Dr. Poranen,

Thank you for submitting your manuscript to PLOS ONE. After careful consideration, we feel that it has merit but does not fully meet PLOS ONE’s publication criteria as it currently stands. Therefore, we invite you to submit a revised version of the manuscript that addresses the points raised during the review process.

Congratulations, Our reviewers have returned favorable comments on your manuscript. The have made some recommendations that I agree will significantly improve your paper, and I am confident that we wil be able to move forward after the reviewer's concerns are addressed. I have included comments below.

Respectfully, David Wampler PhD

We look forward to receiving your revised manuscript.

Kind regards,

David Wampler

Academic Editor

PLOS ONE

Journal Requirements:

Open access funded by Helsinki University Library

4. Please remove all personal information, ensure that the data shared are in accordance with participant consent, and re-upload a fully anonymized data set.

Reviewers' comments:

Reviewer's Responses to Questions

**Comments to the Author**

1. Is the manuscript technically sound, and do the data support the conclusions?

Reviewer #1: Yes

2. Has the statistical analysis been performed appropriately and rigorously? 

Reviewer #1: Yes

3. Have the authors made all data underlying the findings in their manuscript fully available?

Reviewer #1: No

4. Is the manuscript presented in an intelligible fashion and written in standard English?

Reviewer #1: Yes

5. Review Comments to the Author

Reviewer #1: The authors continue their work amidst a deeply needed task: seeking to understand the significant factors that drive or hinder our paramedic colleagues in the field. If we can understand the highest-leverage issues, we’re more likely to serve as a positive force in medical direction, training, or administrative tasks.

Paper flowed well; I enjoyed reading it, and suspect others will also.

Please forgive attempts to note minor editing/formatting issues that would be the purview of the editor(s); I assume they’re quite busy and offer my observations as I see them.

Major comment(s):

Results, page 9 of manuscript, lines 220-226, also discussion, page 23, lines 481-492: (Bottom line up front, please add to strengths/limitations) One of the only criticisms I had about this excellent work pertains to selection bias/healthy volunteer effect/survivorship bias – the fact that the ‘expert’ participants were largely self-selected, and skewed (possibly?) older (acknowledging the requirement to have at least 5 years in EMS, and the fact that one had to have a bachelor’s degree to begin with) means there won’t be as many younger people available to participate – but perhaps more importantly, in Wald statistic fashion, we only know what was most important to 47-ish people willing to speak up about it. Those who left the profession earlier in the career, or those who elected not to enroll, there’s no reason not to think those people wouldn’t prove systematically different to the enrollees. All those words said, ultimately the work reveals themes deemed important by those who remained committed to the career and motivated enough to respond, allowing subsequent lines of inquiry.

General comment: Methods changed in the middle of the work – no adequate consensus manifested, so the threshold was (appropriately) “modified;” it goes like that sometimes. Just ensure the methods section adequately reflects that adjustment; currently it’s something of a surprise in the results section (though alluded to, somewhat, with mention of medians in lines 190-193).

Minor comment(s):

Introduction section, page 3 of manuscript, line 54: Consider Oxford comma following “under time pressure”

Introduction “ “, line 57: Consider comma after “social care services,” alternatively a semicolon, or nothing as so desired

Methods, page 7 of manuscript, line 168: (Feb 2024) seemed out of place – was this when the data was analyzed, or the version number of SPSS used was released? Recommend either “IBM SPSS Statistics for Windows, Version XX.0 (IBM Corp., Armonk, NY)” or swap to the specific release date in parentheticals… assuming of course that I’ve correctly parsed your intent

Discussion, page 21, lines 439-440: I would contend that the risks of violence against EMS personnel are well established (citations follow); while it’s not necessarily in the scope of the paper to drive the point of prevalent violence against our prehospital colleagues, I’d suggest we can move beyond stating the violence risks “may” be common, or merely a feeling – they're quite substantial, at least in places it’s been looked at.

Workplace violence prevention for paramedicine clinicians: Proposed U.S. bill addresses a data collection gap. (2023, April 18). Journal of Emergency Medical Services. https://www.jems.com/ems-management/legal-issues/workplace-violence-prevention-for-paramedicine-clinicians-proposed-u-s-bill-addresses-a-data-collection-gap/ (Salient reference therein: Maguire BJ, Al Amiry A, O’Neill BJ. Abstract 3278 – An examination of injuries and illnesses among paramedicine clinicians; using U.S. Department of Labor data to identify risks and compare risks to other occupational groups. American Public Health Association Conference; 2022; Boston, MA.)

Rahimi, M., Fallah Kharmandar, P., Aghababaeian, H., Sotoudeh, F., Afshari, A., & Hasani-Sharamin, P. (2020). Prevalence of workplace violence types against personnel of emergency medical services (EMS): A systematic review and meta-analysis. Health Promotion Perspectives, 10(1), 28–35. https://doi.org/10.15171/hpp.2020.06

McGuire, S. S., Gazley, B., Crowe, R. P., Bentley, M. A., O’Neil, C., & Koller, C. A. (2024). Workplace violence against emergency medical services (EMS): A prospective 12-month cohort study evaluating prevalence and risk factors within a large, multistate EMS agency. Prehospital Emergency Care. Advance online publication. https://doi.org/10.1080/10903127.2024.2411020

6. PLOS authors have the option to publish the peer review history of their article (what does this mean?). If published, this will include your full peer review and any attached files.

Reviewer #1: **Yes: **Ian L Hudson, DO, MPH

---

## [Author Response · Author response to Decision Letter 1]

19 Aug 2025

Dear Dr. David Wampler,

We were pleased that you gave us an opportunity to submit a revised draft of our manuscript: "The role of human factors in paramedics’ clinical judgement – A modified Delphi study”. We are grateful for the insightful comments provided by you and the reviewer. Those comments dealt with important issues, leading us to revise the manuscript in accordance with those comments.

Please see below for a point-by-point response to the concerns raised by you and the reviewer. We have highlighted revisions in the text in yellow. Line numbers below refer to the marked copy; we have additionally uploaded a clean copy.

We hope that our response adequately addresses all the points raised and we look forward to hearing from you.

Sincerely,

Anna Poranen, MSc, on behalf of all the authors

Comments and concerns:

Our response 1: We have corrected file names and checked that our manuscript meets PLOS ONE’s style requirements.

Open access funded by Helsinki University Library

Our response 2: We have now added the funder’s role in the Financial disclosure -section. It now reads: “Open access was funded by Helsinki University Library. The funder had no role in study design, data collection and analysis, decision to publish, or preparation of the manuscript.”

Our response 3: We have now updated our Data Availability statement as follows:

The datasets generated and/or analysed in the current study are not publicly available because of privacy concerns. The informed consent indicated that the collected data would be used solely for this study, and no prior consent was obtained from the Delphi panelists to share their responses beyond the research team. While the content of the questionnaire and the response options are described in this paper, full transcripts of the open-ended responses have not been provided in order to preserve participant confidentiality. However, data access requests may be made to the corresponding author (anna.e.poranen@helsinki.fi) upon reasonable request, provided that participant confidentiality can be maintained. Alternatively, requests may also be directed to the University of Helsinki Research Ethics Committee in the Humanities and Social and Behavioural Sciences (eettinen-toimikunta@helsinki.fi).

4. Please remove all personal information, ensure that the data shared are in accordance with participant consent, and re-upload a fully anonymized data set.

Our response 4: Please see our previous response.

Our response 5: Thank you for this guideline, we have acted accordingly.

Our response 6: We reviewed our reference list in accordance with the instructions. In addition, we have added two new references, please see Our response 13.

Reviewers' comments:

Reviewer #1: The authors continue their work amidst a deeply needed task: seeking to understand the significant factors that drive or hinder our paramedic colleagues in the field. If we can understand the highest-leverage issues, we’re more likely to serve as a positive force in medical direction, training, or administrative tasks.

Paper flowed well; I enjoyed reading it, and suspect others will also.

Our response 7: Thank you, we appreciate your feedback and time given to our article.

Please forgive attempts to note minor editing/formatting issues that would be the purview of the editor(s); I assume they’re quite busy and offer my observations as I see them.

Major comment(s):

Results, page 9 of manuscript, lines 220-226, also discussion, page 23, lines 481-492: (Bottom line up front, please add to strengths/limitations) One of the only criticisms I had about this excellent work pertains to selection bias/healthy volunteer effect/survivorship bias – the fact that the ‘expert’ participants were largely self-selected, and skewed (possibly?) older (acknowledging the requirement to have at least 5 years in EMS, and the fact that one had to have a bachelor’s degree to begin with) means there won’t be as many younger people available to participate – but perhaps more importantly, in Wald statistic fashion, we only know what was most important to 47-ish people willing to speak up about it. Those who left the profession earlier in the career, or those who elected not to enroll, there’s no reason not to think those people wouldn’t prove systematically different to the enrollees. All those words said, ultimately the work reveals themes deemed important by those who remained committed to the career and motivated enough to respond, allowing subsequent lines of inquiry.

Our response 8: Thank you for this important comment. Prior to data collection, we agreed that a minimum of five years work experience would ensure that experts have sufficient clinical competence of EMS to guarantee the credibility of this study. That is why we chose not to include paramedics who had only recently graduated. However, it is important to acknowledge that there may be potential bias of self-selection.

We have now clarified this in the Strengths and limitations section, and it now reads:

“Another potential bias may be self-selection as participation may have predominantly included paramedics who were particularly motivated to share their views on the topic. This may have influenced the representativeness of the findings.” (Lines 497-500)

General comment: Methods changed in the middle of the work – no adequate consensus manifested, so the threshold was (appropriately) “modified;” it goes like that sometimes. Just ensure the methods section adequately reflects that adjustment; currently it’s something of a surprise in the results section (though alluded to, somewhat, with mention of medians in lines 190-193).

Our response 9: Thank you for this important suggestion. We have now revised the text in the Methods -section. It now reads:

“While the initial plan was to analyse consensus based on percentage agreement, the analysis approach was adjusted after Round 2 and medians and interquartile ranges (IQR) were calculated allowing a more nuanced assessment of consensus in Likert-scale responses.” (Lines 191-193)

Minor comment(s):

Introduction section, page 3 of manuscript, line 54: Consider Oxford comma following “under time pressure”

Our response 10: We have now added an Oxford comma: “Paramedics work in challenging circumstances, under time pressure, and have to treat and deal with a wide range of medical and social problems and make quick and effective decisions.” (Line 54)

Introduction “ “, line 57: Consider comma after “social care services,” alternatively a semicolon, or nothing as so desired

Our response 11: We have now added comma after “social care services” and it now reads:

“In many rapidly ageing societies, including Finland, emergency medical services (EMS) have increasingly started to complement other health and social care services, and paramedics often assess and treat patients in their homes in a more comprehensive way [3, 4].” (Line 57)

References:

3. Ebben RHA, Vloet LCM, Speijers RF, Tönjes NW, Loef J, Pelgrim T, et al. A patient-safety and professional perspective on non-conveyance in ambulance care: a systematic review. Scand J Trauma Resusc Emerg Med. 2017;25(1):71.

4. Paulin J. Non-conveyance and patient safety in prehospital emergency care [Dissertation]: University of Turku; 2022.

Methods, page 7 of manuscript, line 168: (Feb 2024) seemed out of place – was this when the data was analyzed, or the version number of SPSS used was released? Recommend either “IBM SPSS Statistics for Windows, Version XX.0 (IBM Corp., Armonk, NY)” or swap to the specific release date in parentheticals… assuming of course that I’ve correctly parsed your intent

Our response 12: Thank you for pointing this out. We have now adjusted the text to better reflect the intended meaning. It now reads:

“Descriptive statistics were calculated using IBM SPSS Statistics for Windows, version 28.0 (IBM Corp., Armonk NY) and the analysis was conducted in February 2024.” (Lines 168-170)

Discussion, page 21, lines 439-440: I would contend that the risks of violence against EMS personnel are well established (citations follow); while it’s not necessarily in the scope of the paper to drive the point of prevalent violence against our prehospital colleagues, I’d suggest we can move beyond stating the violence risks “may” be common, or merely a feeling – they're quite substantial, at least in places it’s been looked at.

Workplace violence prevention for paramedicine clinicians: Proposed U.S. bill addresses a data collection gap. (2023, April 18). Journal of Emergency Medical Services. https://www.jems.com/ems-management/legal-issues/workplace-violence-prevention-for-paramedicine-clinicians-proposed-u-s-bill-addresses-a-data-collection-gap/ (Salient reference therein: Maguire BJ, Al Amiry A, O’Neill BJ. Abstract 3278 – An examination of injuries and illnesses among paramedicine clinicians; using U.S. Department of Labor data to identify risks and compare risks to other occupational groups. American Public Health Association Conference; 2022; Boston, MA.)

Rahimi, M., Fallah Kharmandar, P., Aghababaeian, H., Sotoudeh, F., Afshari, A., & Hasani-Sharamin, P. (2020). Prevalence of workplace violence types against personnel of emergency medical services (EMS): A systematic review and meta-analysis. Health Promotion Perspectives, 10(1), 28–35. https://doi.org/10.15171/hpp.2020.06

McGuire, S. S., Gazley, B., Crowe, R. P., Bentley, M. A., O’Neil, C., & Koller, C. A. (2024). Workplace violence against emergency medical services (EMS): A prospective 12-month cohort study evaluating prevalence and risk factors within a large, multistate EMS agency. Prehospital Emergency Care. Advance online publication. https://doi.org/10.1080/10903127.2024.2411020

Our response 13: Thank you for this comment, this is important to clarify. We have now revised the text as follows:

“This is in line with previous studies showing that risks to occupational safety and the risk of violence against paramedics are common [35, 36]. For example, systematic review by Murray et al. [35] shows that 57-93% of paramedics have experienced verbal and/or physical violence at least once during their career.” (Lines 443-446)

References:

35. Murray RM, Davis AL, Shepler LJ, Moore-Merrell L, Troup WJ, Allen JA, et al. A Systematic Review of Workplace Violence Against Emergency Medical Services Responders. New Solut. 2020;29(4):487-503.

36. McGuire SS, Bellolio F, Buck BJ, Liedl CP, Stuhr DD, Mullan AF, et al. Workplace Violence Against Emergency Medical Services (EMS): A Prospective 12-Month Cohort Study Evaluating Prevalence and Risk Factors Within a Large, Multistate EMS Agency. Prehospital Emergency Care. 2024(1-8).

---

## [Decision Letter · Decision Letter 1]

29 Aug 2025

The role of human factors in paramedics’ clinical judgement – A modified Delphi study

PONE-D-25-34423R1

Dear Dr. Poranen,

We’re pleased to inform you that your manuscript has been judged scientifically suitable for publication and will be formally accepted for publication once it meets all outstanding technical requirements.

Kind regards,

David Wampler

Academic Editor

PLOS ONE

Additional Editor Comments (optional):

Dear Dr. Anna Poranen

Congratulations, our reviewers have indicated that your work is ready to move forward in the publication process.

I would like to personally thank you for your contribution to PLOS One!

Congrats,

David Wampler, PhD

Academic Editor

Reviewers' comments:

Reviewer's Responses to Questions

**Comments to the Author**

1. If the authors have adequately addressed your comments raised in a previous round of review and you feel that this manuscript is now acceptable for publication, you may indicate that here to bypass the “Comments to the Author” section, enter your conflict of interest statement in the “Confidential to Editor” section, and submit your "Accept" recommendation.

Reviewer #1: All comments have been addressed

2. Is the manuscript technically sound, and do the data support the conclusions?

Reviewer #1: Yes

3. Has the statistical analysis been performed appropriately and rigorously? 

Reviewer #1: Yes

4. Have the authors made all data underlying the findings in their manuscript fully available?

Reviewer #1: No

5. Is the manuscript presented in an intelligible fashion and written in standard English?

Reviewer #1: Yes

6. Review Comments to the Author

Reviewer #1: Overall: As before, an excellent study focused on the perceptions of the end-users, who ultimately do the work and face the uncertainty; understanding what weighs on their ability to execute professionally and remain resilient should concern every conscientious medical director and EMS administration.

Comments were thoughtfully considered and modifications implemented where authors deemed appropriate. Remaining points found are miniscule (and apologies if these were present before and I simply focused on other things):

Phrasing of the sentence in line 110-111, "Both types of units are mainly used for phone-based requests for care instructions by paramedics..." inscrutable to me, though I may simply be slow today. Unclear how a HEMS unit gets use for phone-based requests for care instructions by paramedics; are we discussing that both types of units are summoned for on-site care i.e., treat-and-release/no-load/patient initiated refusal of conveyance calls?

Line 140: "The literature review process is shown in S1 file." Recommend either: a) definite article before S1 i.e., "is shown in [the] S1 file," or b) drop 'file' i.e., "... is shown in S1." or c) spell it out "... is shown in Supplement 1 (S1)." It's small, but it was enough to pull me out of the flow.

Thanks for generating this meaningful work.

7. PLOS authors have the option to publish the peer review history of their article (what does this mean?). If published, this will include your full peer review and any attached files.

Reviewer #1: **Yes: **Ian L. Hudson, DO, MPH

---

## [Editor Report · Acceptance letter]

PONE-D-25-34423R1

PLOS ONE

Dear Dr. Poranen,

I'm pleased to inform you that your manuscript has been deemed suitable for publication in PLOS ONE. Congratulations! Your manuscript is now being handed over to our production team.

Kind regards,

on behalf of

Dr. David Wampler

Academic Editor

PLOS ONE